# A Metabolically Stable Apelin-13 Analog Acting as a Potent I_To_ Potassium Current Blocker with Potential Benefits for Brugada Syndrome

**DOI:** 10.3390/ijms26062735

**Published:** 2025-03-18

**Authors:** Juan Antonio Contreras Vite, Alexandria Tiffinger, Léa Théroux, Nathalie Morin, Mannix Auger-Messier, Pierre-Luc Boudreault, Philippe Sarret, Olivier Lesur, Robert Dumaine

**Affiliations:** Institute of Pharmacology of Sherbrooke, Université de Sherbrooke, Sherbrooke, QC J1H 5N4, Canada; juan.antonio.contreras.vite@usherbrooke.ca (J.A.C.V.); abvesselle@gmail.com (A.T.); lea.theroux@usherbrooke.ca (L.T.); nathalie.morin@usherbrooke.ca (N.M.); mannix.auger-messier@usherbrooke.ca (M.A.-M.); pierre-luc.boudreault@usherbrooke.ca (P.-L.B.); philippe.sarret@usherbrooke.ca (P.S.); olivier.lesur@usherbrooke.ca (O.L.)

**Keywords:** electrophysiology, arrhythmias, sodium channel, potassium channel

## Abstract

Apelin serves as the endogenous ligand for the APJ receptor and enhances cardiac contractility without significantly affecting potassium currents. However, its short in vivo half-life limits clinical application, prompting the development of metabolically stable APJ receptor agonists. This study employed the patch-clamp technique to investigate the effects of the C-terminally modified apelin-13-2Nal derivative (2Nal) on action potential dynamics, rapid sodium (I_Na_), and transient potassium (I_TO_) currents in rat cardiomyocytes. We discovered that 2Nal prolongs ventricular action potential duration by selectively blocking I_To_. Dose-response analysis indicated that 2Nal acts as a partial antagonist of I_TO_, achieving a maximum blockade of 47%, with an apparent EC50 of 0.3 nM, while not affecting I_Na_. Our lab previously found that an imbalance between I_To_ and I_Na_ currents contributes to the development of cardiac arrhythmias in conditions like Brugada syndrome. Currently, few therapeutic options exist to safely address this imbalance, as sodium channel openers cannot restore it, and most I_To_ blockers are cardiotoxic. The selective blockade of I_To_ by 2Nal that we describe here helps restore the balance of electrical currents between I_To_ and I_Na_. Our study presents a novel, safe partial antagonist of I_To_ that may help prevent arrhythmias associated with Brugada syndrome.

## 1. Introduction

Apelin is an endogenous peptide recognized for its therapeutic potential in hypertension and heart failure [1,2,3,4], as well as for its modulation of the endocrine system [5,6,7]. This adipokine is primarily synthesized in adipose tissue but is also produced in the gastrointestinal tract, brain, kidney, lung, mammary glands, and cardiovascular system [8,9]. The interest in apelin arises from its dual vasodilatory effects and strong positive cardiac inotropic effects [10]. Apelin triggers arterial vasodilation by enhancing nitric oxide (NO) production in the endothelium [11,12,13,14,15] and facilitating β-arrestin recruitment [16]. The cardiac inotropic effects are believed to involve modulation of intracellular calcium concentration in cardiomyocytes [4,17,18,19,20] and potentially increasing myofibril sensitivity [20]. Notably, previous studies have indicated that these effects on contractility occur independently of alterations in potassium current amplitudes [19,20].

While limited, some studies have demonstrated that apelin ligands interact with cardiac ion channels. For instance, we found that native apelin peptides modulate cardiac sodium current (I_Na_) gating and enhance its amplitude, which may contribute to increased cardiac excitability [21]. This increase in I_Na_ could play a role in apelin’s inotropic effect by influencing intracellular Na^+^ concentration and sodium–calcium exchanger turnover [21]. Other laboratories have reported that apelin reduces action potential duration (APD) and blocks the ultra-rapid potassium current in rabbit left-atrium cardiomyocytes [22], while also inhibiting large conductance calcium-activated K^+^ channels [23]. However, there is currently no evidence that apelin modulates the voltage-gated calcium current (I_Ca,L_), and previous studies have reported no effects of apelin on the transient outward potassium current, I_To_, and its sustained component, I_Ksus_ [19,20].

Given its inodilator effects, apelin is considered a promising candidate to mitigate some adverse effects of β-blockers and could be beneficial for treating heart failure and hypertension, as well as supporting vasopressors during septic shock [24,25]. However, the predominant isoforms identified in vivo apelin-36, -17, and -13 are highly susceptible to proteolysis [9,26]. For example, angiotensin-converting enzyme 2 (ACE-2) cleaves apelin-13 between the Pro^12^ and Phe^13^ residues, reducing its half-life to less than 1 min in vivo [27]. This short half-life significantly limits the clinical utility of apelin. To enhance its therapeutic potential, synthetic forms of apelin with longer half-lives are needed, leading to the development of various apelin analogues [28,29,30,31,32,33,34,35].

We discovered that macrocyclization of apelin-13 improves its binding properties and plasma stability [33,34,35]. Our goal in this study was to determine if this new compound could alter cardiac electrophysiology. Given the known inotropic effects of Apelin, we tested the effect of 2Nal on the sodium and potassium current I_To_ in rat cardiomyocytes. In this paper, we present data demonstrating that the C-terminally modified analog, 2Nal, where the terminal Phe of apelin-13 is replaced by the unnatural amino acid 2Nal, serves as a potent partial blocker of I_To_ with minimal impact on I_Na_. This characteristic makes 2Nal an attractive candidate for treating Brugada syndrome (BrS). One proposed mechanism for arrhythmias in BrS suggests that mutations in the cardiac sodium channel cause a reduction in fast inward sodium current (I_Na_). This reduction in I_Na_ amplitude alters the balance of current between I_Na_ and I_To_ current, thus leading to a stronger repolarizing force by I_To_ and premature repolarization of the ventricular action potential [36]. We hypothesize that selective blockade of I_To_ by 2Nal may restore the balance of current and prevent premature repolarization, thus offering potential therapeutic benefits for the treatment and prevention of BrS.

## 2. Results

We first examined the effects of 2Nal on the AP of ventricular cardiomyocytes (Figure 1A). Perfusing cells with 5 nM of 2Nal provoked a significant increase in APD from 7.7 ± 0.3 ms to 9.7 ± 0.3 ms (20%), 10.9 ± 0.6 ms to 15.1 ± 1.2 ms (28.5%), and 16.5 ± 1.3 ms to 32 ± 2.2 ms (48.5%) at 30%, 50%, and 90% of repolarization, respectively (Figure 1B). The largest increase in APD was observed between 50% and 90% repolarization of the cell membrane potential, suggesting a reduction in potassium currents participating in final repolarization.

In rat ventricular cardiomyocytes, the final repolarization of the action potential is mainly governed by the sustained component of I_To_ (I_To,sus_). We therefore tested if blockade of I_To_ peak current (I_To,peak_) and I_To,sus_ by 2Nal contributed to AP prolongation. Figure 2 shows that 2Nal blocked the I_To_ current. The current–voltage (I/V) relationships (Figure 2B) show that blocking of I_To,peak_ and I_To,sus_ by 2Nal increased in a dose-dependent manner. However, I_To,sus_ blockade was more abrupt, suggesting a higher affinity for 2NaL. In both cases, blockade by 2Nal saturated at a concentration of 100 nM.

Analysis of the dose–response curves for I_To,peak_ and I_To,sus_ (Figure 3) revealed a block that could be fitted to a single binding kinetics equation and saturated for 2Nal concentrations above 10 nM (Figure 3A). Maximum blocking reached 34.2 ± 5.6% and 47 ± 9.2% of I_To,peak_ and I_To,sus_, respectively (Figure 3C). Therefore, 2Nal is a partial antagonist of I_To_. Half-maximal blocking concentrations calculated from the Hill equation fit to data indicate that I_To,sus_ is 10X more sensitive to 2Nal, with IC_50_ values of 0.024 ± 0.01 nM vs. 0.3 ± 0.1 nM for I_To,peak_ (Figure 3B).

Blockade of I_To_ may be caused by direct steric hindrance of 2Nal in the pore of potassium channels or by alterations in their availability at various membrane voltages (inactivation). To evaluate the influence of the latter, we next tested the effect of 2Nal (5 nM and 100 nM) on I_To_ steady-state inactivation (Figure 4). Despite a slight trend in the top steady-state inactivation curves toward hyperpolarized potentials, the mid-inactivation voltage values (V_h_) were not statistically different between the control (ctrl) and 2Nal groups (Table 1).

Arrhythmias associated to BrS may be explained by an imbalance between the inward sodium current I_Na_ and I_To_ promoting more rapid early repolarization of the cardiac ventricular epicardium [36,37]. Blocking I_To_ may prevent early repolarization if it does not block its I_Na_ counterpart. We therefore tested if 2Nal blocked I_Na_, as previously reported for apelin-13 [21]. Figure 5 shows that 2Nal did not affect I_Na_ peak amplitude, its current–voltage relationship (Figure 5D,E). However, 2Nal slightly hyperpolarized I_Na_ activation, with the mid-activation voltage (V_0_._5_) shifting from −42 ± 1 mV to −46 ± 2.3 mV (Figure 5C).

Availability (steady-state inactivation) of Na^+^ channels at various resting membrane voltages determines the amplitude of I_Na_ opposing I_To_ during an action potential. Therefore, we tested the effect of 2Nal on I_Na_ steady-state inactivation (Figure 6). Our results show that 2Nal shifted steady-state inactivation by 5 mV, from a mid-inactivation voltage of −74.6 ± 0.6 mV for control (ctrl) to −80.2 ± 2 mV (Figure 6B). However, neither shift in steady-state activation or inactivation was sufficient to significantly alter the maximum conductance (G_Na,Max,_) of I_Na_ (Figure 5E).

In rats, I_To_ is generated by tetramers formed by the molecular correlates Kv4.2 and Kv4.3 [38,39], whereas in humans, the potassium channel Kv4.3 is dominant [39]. To confirm that 2NaL also interact with Kv4.3, we tested its effect in TSA201 (a variant of HEK cells) transfected with the human isoform hKv4.3. Figure 7 shows that 2Nal blocked 20 ± 4% and 25 ± 6% of the current generated by hKv4.3, confirming that 2Nal should also block I_To_ in human cardiomyocytes.

## 3. Discussion

In this study, we tested the effect of 2Nal, a C-terminally modified apelin-13 derivative synthesized by our group [28], on cardiac action potential (AP), I_To_, and I_Na_ of rat ventricular myocytes. The interest of our study lies in the fact that, unlike apelin [19,20], 2Nal blocks I_To_. Moreover, blocking occurs at concentrations within a therapeutic range and saturates at approximately 40% of the maximum amplitude of I_To_. Our data also indicate that 2Nal does not block the cardiac sodium current directly opposing I_To_ and has minimal effect on its availability at normal resting potential of cardiomyocytes. Those features suggest that selective blockade of I_To_ by 2Nal potentiates the depolarizing power of I_Na_. This may in part explain the prolongation of the ventricular APD we observed in rats (Figure 1). However, in larger species, such as humans, where the APD is longer, the imbalance between I_To_ and I_Na_ will have more of an impact during the early phase 1 (notch) of the AP and less during late repolarization, where other currents are activated. Disruption of the balance of current between I_Na_ and I_To_, especially in the epicardial layer of the ventricle, is one of the primary mechanisms proposed to explain BrS.

BrS has been associated with reduced I_Na_ and loss-of-function mutations in the cardiac sodium channel gene Scn5a. The latter can lead to reduced expression or expression of non-functional proteins [36,40,41,42,43,44]. Reduced amplitude of I_Na_ will potentiate early repolarization by I_To_ [45,46,47,48]. In the worst case, cardiac regions where I_To_ is most prominent, such as the right ventricular epicardium, will repolarize within milliseconds, while the inner part of the ventricle will remain fully depolarized for hundreds of milliseconds. This dispersion of repolarization within the cardiac ventricle may trigger shunt currents from depolarized areas to the fully repolarized regions. Consequently, fully repolarized regions of the epicardium may be prematurely re-excited and trigger re-entry arrhythmias. Restoration of the current balance between I_Na_ and I_To_ is viewed as a therapeutic strategy to prevent BrS. This can be achieved by blocking I_To_ or by increasing I_Na_. Our data showing specific blockade of I_To_ suggest that 2Nal has the potential to restore the balance of current needed to prevent arrhythmias in BrS. Moreover, we found that 2Nal acts as a partial antagonist of potassium channels by blocking a maximum of only 46% of I_To_. This is a highly desirable property for a potassium current blocker, as it significantly increases the margin of safety in case of overdose. Therefore, our data suggest that 2Nal may be an alternative treatment to the currently more invasive cardiac defibrillator implantation used for BrS.

In rat cardiomyocytes, 2Nal at a concentration 10X higher than the one blocking I_To_ had no effect on I_Na_ amplitude. While 2Nal at 5 nM altered the steady-state activation and inactivation, the changes were not sufficient to alter the maximum peak current of I_Na_. However, at this high concentration, we observed a small slowing of the inactivation kinetics of I_Na_ (Figure 5A) that may provide a small depolarizing force and explain the small kink observed in the shape of phase 1 repolarization of the action potential (Figure 1A). Since most of the AP prolongation occurs at 90% repolarization, at a time when I_Na_ is completely inactivated, we conclude that the APD prolongation observed is primarily due to blockade of I_To_.

IWe observed, the onset of I_To_ blockade by 2Nal occurred within 3 min. This rapid effect suggests a direct interaction of 2Nal with the molecular correlates of I_To_ rather than activation of the APJ receptors. This hypothesis is further supported by our previous data showing that activation of the APJ receptors by apelin induce an increase in I_Na_ [21]. At this stage, we cannot distinguish how specific affinities of Kv4.2 or Kv4.3 for 2Nal modulate the overall effect on I_To_ amplitude in native cardiomyocytes. However, our observation of apparent EC_50_ differing between the peak and sustained components of I_To_ suggests that both channels are sensitive to 2Nal. The apparent affinity of human I_To_ for 2Nal remains to be determined, but our data in TSA201 cells show that 2NaL blocks the current generated by hKv4.3 and suggest that the effect observed in rats also occurs in humans.

## 4. Materials and Methods

### 4.1. Cell Isolation 

Ventricular cardiomyocytes from adult rats were isolated by enzymatic dissociation as previously reported [21]. Adult Sprague Dawley rats (Charles River, Montreal, QC, Canada) were heparinized (5000 U) and sedated with 2% Isoflurane in an oxygen chamber (2 L/min). Hearts were excised by thoracotomy and cannulated via the aorta to be mounted on a Langendorff perfusion apparatus. Hearts were initially perfused with standard calcium-free Tyrode solution [21] at 37 °C (3–5 min), at a rate of 13 mL/min, and pre-digested by perfusion with low-calcium (150 μM) Tyrode solution supplemented with 220–280 U/mg of low-trypsin collagenase type 2 (Worthington Biochemical, Toronto, ON, Canada) for less than 15 min.

Once removed from the Langendorf perfusion apparatus, the heart septum and left ventricle were cut and gently shaken for 3 min in a 50/50 solution (digestion sln/low-calcium Tyrode solution) to disperse the isolated cells. The supernatant of the cell solution was filtered, and cells were left to settle in a 15 mL conical tube. The procedure was repeated 4 times, and cells were divided into fractions. Cardiomyocyte pellets from each fraction were washed with low-calcium Tyrode solution. For the final washing, supernatant solution was removed and replaced by standard Kraftbrühe (KB) solution [21], and the cells were kept at 4 °C.

### 4.2. Animal Care

Maintenance, care, and all animal protocols followed the ARRIVE (Animals Research: Reporting of In vivo Experiments) guidelines approved by the Institutional Animal Care and Use Committee of the Faculty of Medicine and Health Sciences of the Université de Sherbrooke (protocol no. 2021-3039).

### 4.3. Cell Culture and Transfection

TSA201 cells were transfected with 9 μg of pcDNA3.1/Kv4.3-short plasmid containing the short sequence encoding human cardiac Kv4.3 (hKv4.3). Transfection was performed using lipofectamine (ratio DNA/Lipo, 1:2) according to the manufacturer’s instructions (GIBCO, Invitrogen, Waltham, MA, USA). Cells were cultured in Dulbecco’s Modified Eagle’s Medium (DMEM) supplemented with 10% fetal bovine serum, 1%glutamine, and 1% penicillin/streptomycin at 37 °C, in a 95% O_2_ and 5% CO_2_ atmosphere.

### 4.4. Compound Synthesis

The C-terminal Phe^13^ residue of apelin-13 was replaced by (2-naphtyl)-L-alanine (2Nal). For ease of reading, we will refer to this compound as 2Nal throughout the article. The synthesis was performed at the 0.1 mmol scale using Fmoc-based chemistry, as described previously [49]. Briefly, the Fmoc N-protected 2Nal residue was attached to the 2-chlorotrityl chloride resin (loading of 0.25 mmol/g) in the presence of *N,N*-diisopropylethylamine (DIPEA) in dichloromethane (DCM). After capping the unreacted 2-chlorotrityl chloride groups with a mixture of DCM/MeOH/DIPEA (7/2/1), the Fmoc group was then cleaved with 20% piperidine/*N,N*-dimethylformamide. The following Fmoc N-protected residue was then coupled with [O-(7-azabenzotriazol-1-yl)-1,1,3,3-tetramethyluronium hexafluorophosphate] (HATU) and DIPEA in DMF. Fmoc deprotections and coupling steps were repeated until the full peptide sequence was obtained. The peptide was cleaved from the resin, and the side chains were simultaneously deprotected with a mixture of TFA (trifluoroacetic acid)/H_2_O/TIPS (triisopropylsilane)/1,2-ethanedithiol (EDT) (92.5/2.5/2.5/2.5). The crude product was precipitated in *tert*-butyl methyl ether (TBME) and then purified on the HPLC-MS system from Waters (Milford, USA) (column XSELECT^TM^ CSH^TM^ Prep C18 (19 × 100 mm) packed with 5 µm particles, UV detector 2998, MS SQ Detector 2, Sample manager 2767, and a binary gradient module) using a binary solvent system (acetonitrile/water + 0.1% formic acid). The desired compound was characterized, and purity (>95%) was assessed using the UPLC/MS system from Waters (Milford, Sunnyvale, CA, USA), using an Acquity UPLC^®^ CSH™ C18 column (2.1 × 50 mm) packed with 1.7 µm particles.

### 4.5. Electrophysiology Recordings

I_Na_ and I_To_ were recorded from freshly isolated left ventricular cardiomyocytes and TSA201 cells after 48 h of transfection at room temperature. Electrical currents were recorded using the patch-clamp technique in whole-cell configuration via an Axopatch 200B amplifier (Molecular Devices, Sunnyvale, CA, USA). Cardiomyocytes and TSA201 cells were pre-incubated with 2Nal during 15 min and continuously perfused during recordings. In preliminary experiments, we measured the onset of the block by 2Nal by initially superfusing with control solution and subsequently washing in 2Nal solution. Recordings were acquired at 10 kHz and filtered to 5 kHz (lowpass Bessel filter). Whole-cell capacitance and series resistance compensation (85%) were optimized to reduce the capacitive artifact and minimize voltage error. Analysis of the recordings was performed using the Clampex 10.7 analysis software.

For Action Potential (AP) recordings, current clamp mode (I = 0) was set, and action potentials were triggered by 1 ms pulses of 0.1 nA current at a frequency of 0.1 Hz.

### 4.6. Electrophysiology Solutions

Extracellular recording solutions for I_Na_ contained (in mM) 117.5 choline-Cl, 10 NaOH, 2.8 Na-acetate, 4 KOH, 1 CaCl_2_, 1.5 MgCl_2_, 20 HEPES, 1 CoCl_2_, 5 TEA, 2 4-AP, 5 BaCl_2_, and 10 glucose. The pH was adjusted with NaOH to 7.4, and osmolarity was adjusted to 295–300 mOsm. Pipette solution contained (in mM) 40 NaOH, 5 NaCl, 5 CsF, 2 MgCl_2_, 10 EGTA, 20 HEPES, 120 cesium aspartate, 0.5 GTP, 3 creatine phosphate, and 2 ATP-Mg. The pH was adjusted with CsOH to 7.3, and osmolarity was adjusted to 295–300 mOsm. The pH adjustment altered the final Na+ content such that the sodium gradient yielded a reversal potential of 0 mV for I_Na_.

Extracellular solution for I_To_ recordings in cardiomyocytes contained (in mM) 126 NaCl, 5.4 KCl, 1 MgCl_2_, 20 HEPES, 2 CaCl_2_, and 11 glucose. The pH and osmolarity were adjusted to 7.4 and 295–300 mOsm, respectively, using NaOH and sucrose. Pipette solution contained (in mM) 90 K-aspartate, 30 KCl, 1 MgCl_2_, 20 HEPES, 5 NaCl, 5 MgATP, 5.5 Glucose. The pH was adjusted with KOH to 7.3 and osmolarity to 295–300 mOsm. For I_To_ recordings in TSA201 cells, extracellular solution contained 130 NaCl, 5 KCl, 1 MgCl_2_, 2.8 Na-acetate, 10 HEPES, 1.8 CaCl_2_, and 10 glucose. The pH and osmolarity were adjusted to 7.3 and 295–300 mOsm, respectively, using NaOH and sucrose. Pipette solution contained (in mM) 125 K-aspartate, 10 KCl, 1 MgCl_2_, 5 HEPES, 10 NaCl, and 5 MgATP; the pH was adjusted with KOH to 7.2, and osmolarity was adjusted to 295–300 mOsm.

For AP recordings, cardiomyocytes were exposed to bath solution containing (in mM) 126 NaCl, 5.4 KCl, 2 CaCl_2_, 1 MgCl_2_, 20 HEPES, and 11 glucose, and for pipette solution, 90 K-aspartate, 30 KCl, 10 NaCl, 5.5 glucose, 1 MgCl_2_, 10 EGTA, 4 Na-ATP, and 20 HEPES. Extracellular and intracellular solutions’ pH was adjusted to 7.4 (NaOH) and 7.2, respectively (KOH). Recording pipettes were pulled from 1.5 mm O.D. and 1.16 mm I.D. capillary glass (PGT150T-7.5 Harvard Apparatus, Holliston, MA, USA) with a Narishige PP-83 vertical Puller and had resistance between 1.8 and 2.5 MΩ.

### 4.7. Materials

2Nal powder was dissolved in Milli-Q water to make a stock solution of 1 M from which we diluted in extracellular solution to reach the final concentrations.

### 4.8. Data Analysis and Protocols

#### 4.8.1. AP Recordings

Cell-membrane resting potential was kept at −80 mV through current injection in current-clamp mode. APs were triggered by 1 ms pulses of current applied in increments of 0.1 nA, at a frequency of 0.1 Hz, until threshold was reached. Action potential duration (APD) was calculated as the time needed for the AP to drop from its maximum voltage amplitude by 30, 50, and 90% during repolarization. The duration of the AP was measured using the Clampex software.

#### 4.8.2. Voltage-Dependent I_Na_ Channels Activation

I_Na_ was activated by 30 ms voltage membrane (Vm) steps between −80 and 10 mV, in 5 mV increments, from a −120 mV holding potential. Average peak current values were divided by the membrane capacitance (Cm) and plotted against the testing voltage to construct I_Na_ current density relationships (pA/pF). Sodium conductance values (G_Na_) were estimated from the ratio I_Na_/(Vm-E_Na_), where E_Na_ is the sodium reversal potential. Maximum conductance (G_Na,Max_) was obtained from the slope of the linear part fit of the I/V for values more positive than −25 mV. Normalized I_Na_-activation curves were constructed with the ratio G_Na_/G_Na,Max_. Data points were fitted with Boltzmann distribution (Equation (1)) to estimate the mid-activation voltage, V_0_._5_:(1)GNaGNa,Max=11+ezFVm−V0.5RT
where z, F, R, and T are the gating charge, the Faraday constant, the universal constant of gases, and the temperature, respectively

#### 4.8.3. I_Na_ Steady-State Inactivation

I_Na_ was elicited by −30 mV test pulse of 20 ms following 500 ms conditioning potentials from −120 to −20 mV, in 10 mV increments. Inactivation curves were obtained from the ratio I_Na_/I_Na,Max_ for each conditioning potential. Data were fitted with Boltzmann distribution (1) to estimate the mid-inactivation voltage, V_h_.

#### 4.8.4. Voltage-Dependent I_To_ Channels’ Activation

I_To_ was activated by 500 ms voltage membrane (Vm) steps between −30 and 50 mV, in 5 mV increments, from a −80 mV holding potential. Average peak and sustained (non-inactivated) current values were divided by the membrane capacitance (Cm) and plotted against the testing voltage to construct the I_To_ current density relationships (pA/pF). Maximum conductance (G_Max_,_[X]_) was obtained from the slope of a linear regression fit of the I/V to each 2Nal concentration tested.

#### 4.8.5. I_To_ Steady-State Inactivation

I_To_ was elicited by 30 mV test pulse of 500 ms following 2.5 s conditioning potentials from −100 to 0 mV, in 10 mV increments. Inactivation curves were obtained from the ratio I_To_/I_To,Max_ for each conditioning potential.

### 4.9. Dose–Response Curves

The values of the maximum conductance obtained for each drug concentration (G_Max,[X]_) were divided by the average of the maximum conductance obtained under control conditions (G_Max,[ctrl]_), and then the data were plotted as a function 2Nal concentration, and the IC_50_ values (50% blockade) were calculated from the fit of a modified Hill equation to data:(2)$$\frac{G_{Max,[X]}}{G_{Max,[ctrl]}}=1+\frac{f-1}{1+{\left(\frac{{IC}_{50}}{\left[2Nal\right]}\right)}^{n}}$$
where f and n are the fraction of 2Nal-sensitive channels and the cooperative sites, respectively.

### 4.10. Statistics

Experimental data points are presented as data ± SEM, and the number of cells is indicated on each figure. Since our data follow a Poisson distribution, statistics was performed using a one-way ANOVA and Tukey test to compare control and test values.

All experimental work was conducted in accordance with national and international guidelines. The protocol for this study was approved by the Animal Care & Welfare Committee of the College of Animal Science & Technology at Guangxi University (Approval Code: GXU2019-021).

## 5. Conclusions

We present data on a novel partial antagonist molecule with high affinity for I_To_. These pharmacological properties make 2Nal a potentially useful compound for selectively blocking I_To_ and preventing premature ventricular AP repolarization in various arrhythmogenic settings. Because 2Nal does not block I_Na_ and is a partial antagonist of I_To_, it also represents a promising therapeutic approach for prevention of arrhythmias in early repolarization disorders such as BrS.

## Figures and Tables

**Figure 1 ijms-26-02735-f001:**
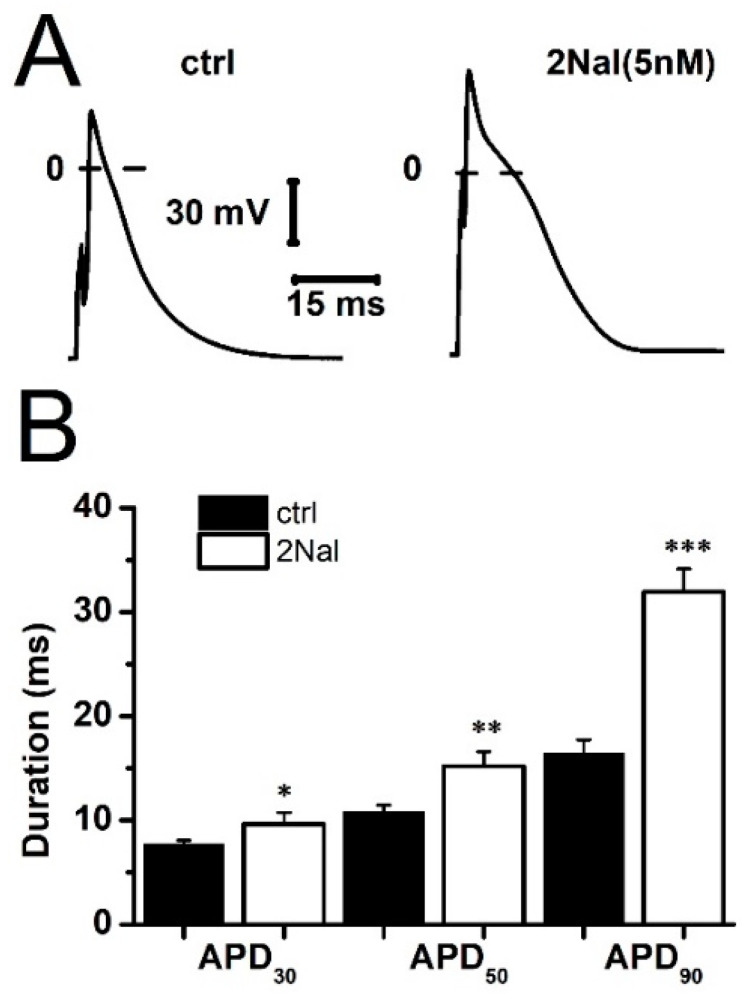
2Nal increases cardiac action potential duration. (**A**) Typical cardiac action potential recordings from rat’s left ventricle cardiomyocytes under control conditions and after perfusion with 2Nal. (**B**) Action potential duration at 30% (APD_30_), 50% (APD_50_), and 90% (APD_90_) of repolarization. Tukey test using ANOVA (ctrl vs. 2Nal). * *p* < 0.05, ** *p* < 0.01, and *** *p* < 0.001. Bars represent average values ± SEM. (*n* = 25, 5 animals).

**Figure 2 ijms-26-02735-f002:**
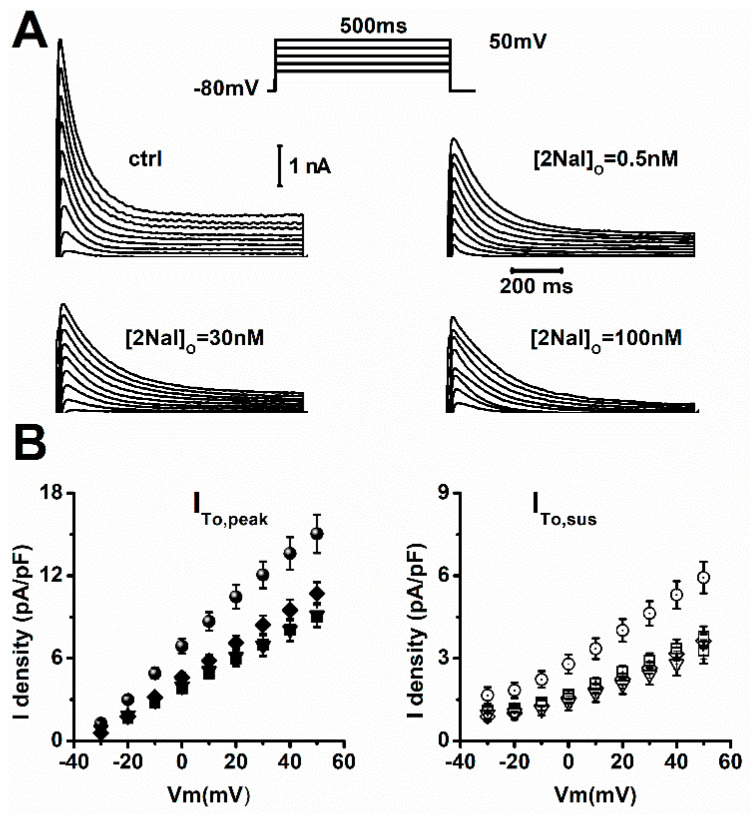
Blockade of I_To_ by 2Nal saturates at 100 nM. (**A**) Representative I_To_ recordings from rat cardiomyocytes exposed to indicated 2Nal concentrations. Currents were activated by 500 ms voltage steps from −30 to 50 mV, in 10 mV increments. Holding potential was −80 mV (protocol inset). (**B**) Current–voltage (IV) relations for peak (I_To,peak_) and sustained (I_To,sus_) components expressed as current density (pA/pF) for control (circles), 0.5 nM (diamonds), 5 nM (squares), and 100 nM (triangles) of 2Nal (*n* = 25, 5 animals).

**Figure 3 ijms-26-02735-f003:**
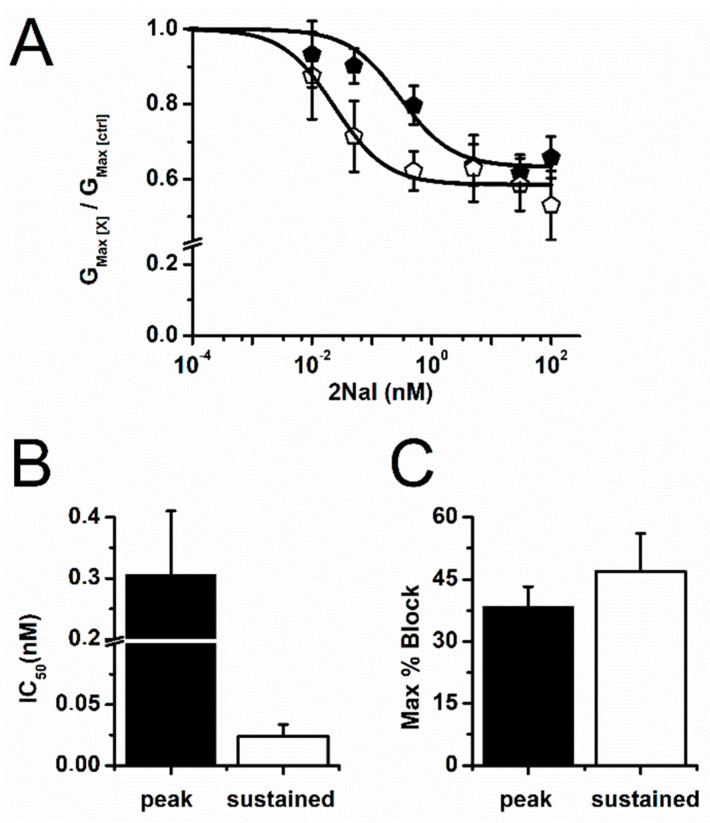
2Nal blocks I_To,sus_ more potently. (**A**) Concentration-dependence curves for I_To,peak_ (filled) and I_To,sus_ (open). The maximum conductance was determined from a linear fit to IV relations for each 2Nal concentration (G_max,[X]_), and then G_max,[X]_ values were divided by the maximum conductance obtained from control IV experiments (G_max,[ctrl]_) and plotted as a function of 2Nal concentration. (**B**) IC_50_ values for I_To,peak_ and I_To,sus_ were estimated from Hill equation fit (solid lines) to the data. (**C**) Maximum I_To_ block (in percent). Bars represent average values ± SEM. (*n* = 32, 6 animals).

**Figure 4 ijms-26-02735-f004:**
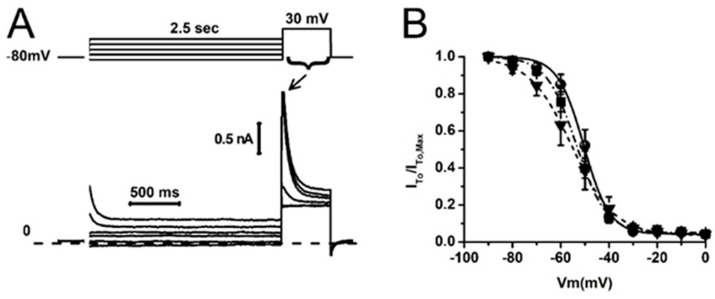
2Nal does not influence I_To_ steady-state inactivation. (**A**) Representative I_To_ recordings during a standard inactivation protocol (top). The currents were activated by a 500 ms test pulse to 30 mV preceded by 2.5 s conditioning test potentials from −100 to 0 mV, in 10 mV increments. (**B**) Steady-state inactivation curves in control conditions (circles), 5 nM (squares), and 100 nM (triangles) of 2Nal. Curves were obtained from the ratio I_To_/I_To,Max_ for each conditioning potentials. Data were fitted with a Boltzmann distribution (overlapping lines). Tukey test using ANOVA showed no significant difference between the mid-inactivation voltage values (ctrl vs. 2Nal); see Table 1 (*n* = 32, 6 animals).

**Figure 5 ijms-26-02735-f005:**
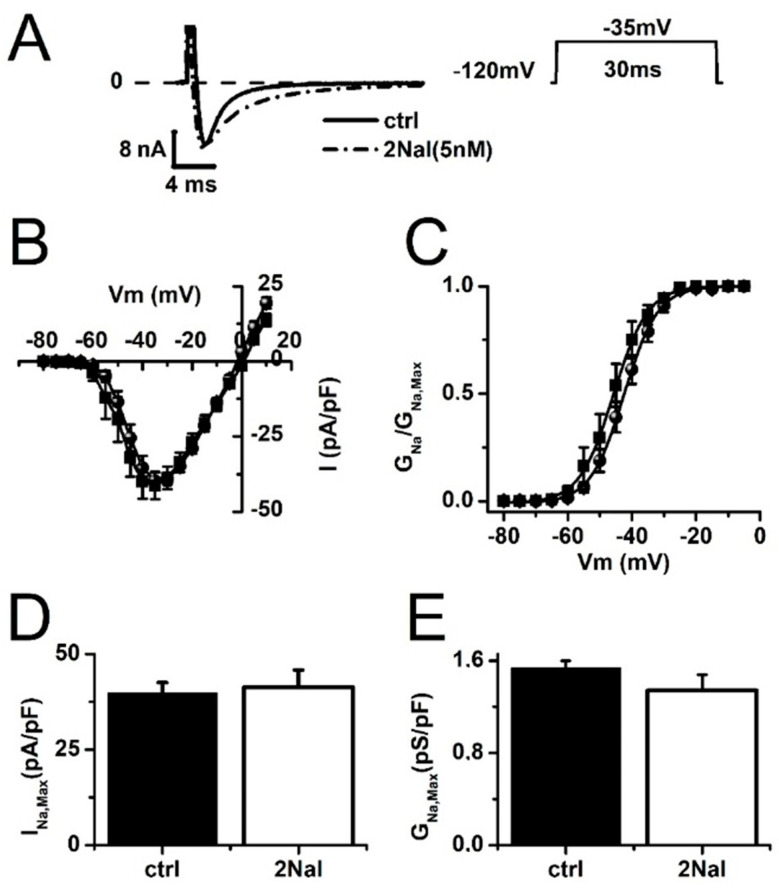
2Nal does not alter peak I_Na_. (**A**) Representative I_Na_ recordings obtained from rat ventricular cardiomyocytes in control conditions and after 15 min incubation with 2Nal. Currents were activated by 30 ms voltage steps from −80 to 10 mV, in 5 mV increments, with a holding potential of −120 mV. (**B**) Current–voltage relations curves expressed as current density for control (circles) and for 5 nM of 2Nal (squares). (**C**) I_Na_ activation curves (G_Na_/G_Na,Max_) for control (circles) and for 5 nM of 2Nal (squares). Activation curves were constructed with I_Na_ conductance (G_Na_) values calculated from the ratio I_Na_/(Vm-E_Na_), and then G_Na_ values were divided by the maximum I_Na_ conductance (G_Na,Max_), which was estimated from the linear part of IV relations more positive than −25 mV. E_Na_ is the sodium reversal potential. (**D**,**E**) Maximum value of I_Na_ (I_Na,Max_) and G_Na,Max_ for control and 5 nM of 2Nal. Tukey test using one-way ANOVA showed no significant difference for the values of I_Na,Max_ and G_Na,Max_ (ctrl vs. 2Nal). Bars represent average values ± SEM. (*n* = 15, 3 animals).

**Figure 6 ijms-26-02735-f006:**
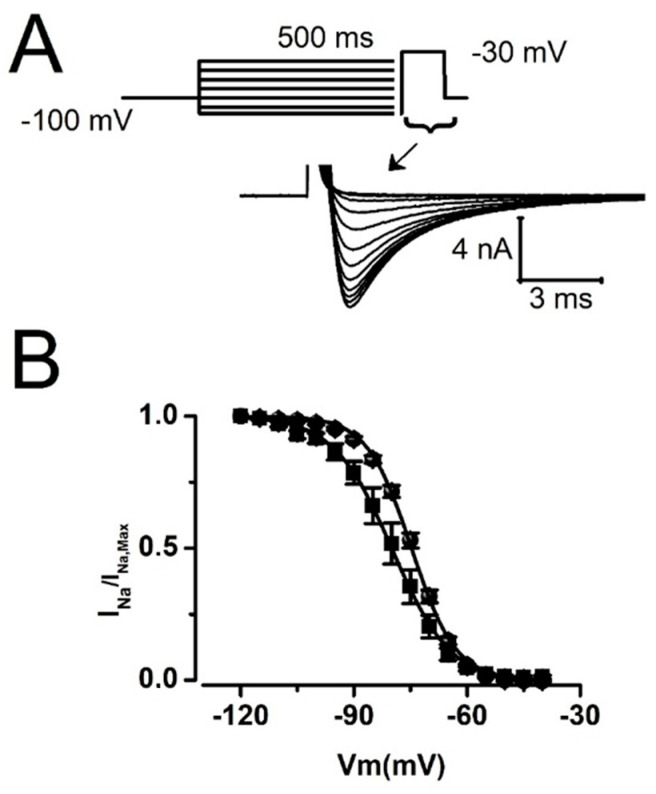
2Nal hyperpolarized I_Na_ steady-state inactivation. (**A**) Representative I_Na_ recorded with a standard inactivation protocol (top). The currents were activated by a 30 ms test pulse to −30 mV following a 500 ms conditioning potential from −120 to −20 mV in 10 mV increments. (**B**) Normalized steady-state inactivation curves in control conditions (circles) and for 5 nM of 2Nal (squares). Curves were obtained from the ratio I_Na_/I_Na,Max_ for each conditioning potentials. Data were fitted with a Boltzmann distribution (overlapping lines; *n* = 25, 3 animals).

**Figure 7 ijms-26-02735-f007:**
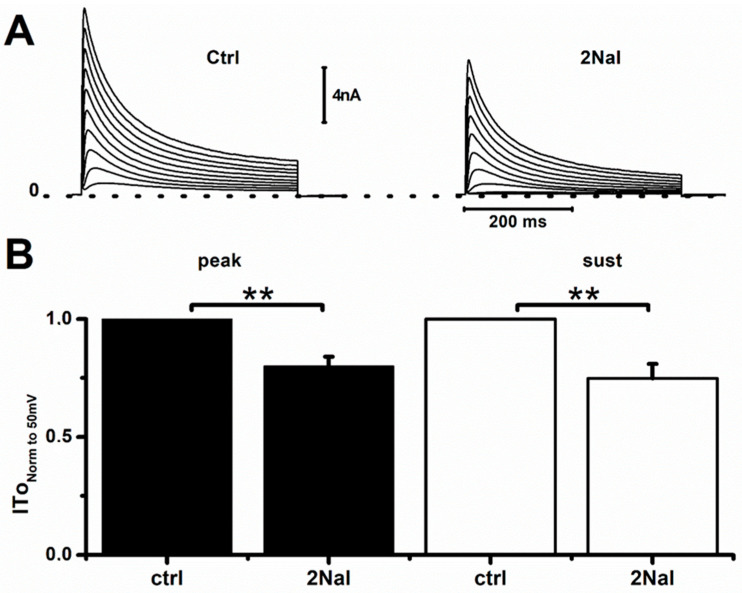
2NaL partially blocked hKv4.3 expressed in TSA201 cells. (**A**) Representative I_To_ recordings from TSA201 cells transfected with hKv4.3 channels, both in control conditions (ctrl) and in presence of 100 nM of 2NaL. (**B**) Normalized values of I_To_ at 50 mV. For each cell, I_To,ctrl_/I_To,2Nal_ ratios were determined and averaged. Tukey test with ANOVA (ctrl vs. 2Nal). ** *p* < 0.01. Bars represent average values ± SEM (*n* = 9).

**Table 1 ijms-26-02735-t001:** I_To_ channels’ mid-inactivation voltage (V_h_) for control conditions and during exposure to 2Nal. Tukey test using ANOVA (ctrl vs. 2Nal) showed no significance difference between V_h_ values (*n* = 21, 5 animals).

2NaL (nM)	V_h_ (mV) ± SEM
0	−51 ± 2
0.05	−48 ± 2
0.5	−49 ± 2
5	−54 ± 2
30	−53 ± 3
100	−56 ± 4

## Data Availability

The data underlying this article will be shared upon reasonable request to the corresponding author.

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
