# Peer review of "A Metabolically Stable Apelin-13 Analog Acting as a Potent ITo Potassium Current Blocker with Potential Benefits for Brugada Syndrome"

_ijms, 2025, doi:10.3390/ijms26062735_

Round 1
Reviewer 1 Report
Comments and Suggestions for Authors
In this study, the authors investigated the effect of 2Nal, a Cter modified apelin, on rat ventricular cardiomyocytes action potential duration, and volage dependent sodium current and potassium current using patch-clamp recording. They showed that 2Nal increased the AP duration at 30, 50 and 90 % repolarization. They also showed that the voltage-dependent K current density Ito is reduced after application of 2Nal. The authors finally show that Kv4.3, the main subunit responsible for Ito in human, expressed in a heterologous system is also partially sensitive to 2Nal.
The results presented in this study are promising and 2Nal represents an appealing avenue to treat cardiac arrhythmias in patients with Brugada syndrome.
However, certain aspects require further clarification:
* N numbers are missing for data presented in Figure 1 to Figure 6. This omission makes the interpretation of statistical analysis difficult.
* Figure 1A, the amplitude of the action potential looks bigger after 2Nal treatment. Also, in Figure 5, the inactivation of INa appears affected by 2Nal treatment. The inactivation parameters should be further analyzed before and after treatment and this needs to be addressed to be able to conclude that 2Nal has no effect on INa. Can a slowing of INa inactivation be responsible for an increase of the amplitude of the AP?
* In discussion, line 346: “2Nal had no effect on INa but blocked ITo after only 3 min of perfusion”, in the Methods section is says “Cardiomyocytes were pre-incubated with 2Nal during 15 minutes…”, this point has to be clarified, especially if the idea is that 2Nal has a direct action on Ito.
Author Response
* N numbers are missing for data presented in Figure 1 to Figure 6. This omission makes the interpretation of statistical analysis difficult.
We thank this reviewer for mentioning it. We added the information to the figure legends
* Figure 1A, the amplitude of the action potential looks bigger after 2Nal treatment.
On average the action potential amplitudes were not significantly different with 85.3 mV and 88.8 mV in control and after application of 2Nal (n=25, 5 animals). However peak AP amplitude reached a slightly more depolarized value because blockade of ITo results in a balance of current favoring INa and a stronger depolarization during the initial phase 0 of the action potential. The 0mV line on the figure gives the impression of a larger AP because of this more depolarized peak but one should notice that the threshold of the AP is also depolarized by 5 mV.
Also, in Figure 5, the inactivation of INa appears affected by 2Nal treatment. The inactivation parameters should be further analyzed before and after treatment and this needs to be addressed to be able to conclude that 2Nal has no effect on INa. Can a slowing of INa inactivation be responsible for an increase of the amplitude of the AP?
Indeed, the slowing of INa may contribute a little and this may explain the kink seen at the end of phase 1 seen in panel A under 2Na. However, we feel that since the amplitude of INa is not altered it will minimally influence AP amplitude. Most of the effect of this slower inactivation will be seen during the early phase 2 (plateau) of the AP. However, total inactivation of INa remains rapid. Since the main effect of 2Nal on APD occurs at 90% repolarization (panel B), at a time when INa is completely inactivated, we feel the influence on ITo remains the dominant effect. We added text at lines 308-312 to address this reviewers concern.
Please note however that we do not wish to enter in an extensive analysis of INa kinetics since the effect are observed occurred at concentrations more than 10x higher (5nM) than the IC50 we observed for ITo (0,3 nM) and will not add to the story, perhaps even confusing the reader.
* In discussion, line 346: “2Nal had no effect on INa but blocked ITo after only 3 min of perfusion”, in the Methods section is says “Cardiomyocytes were pre-incubated with 2Nal during 15 minutes…”, this point has to be clarified, especially if the idea is that 2Nal has a direct action on Ito.
Indeed, this is confusing. Cells were preincubated to shorten the duration under patch-clamp, but perfusion remained through the entire experiment. However, we did a series of control experiments (not shown) to check the onset of the block. We corrected the discussion section accordingly.
Reviewer 2 Report
Comments and Suggestions for Authors
The present study is concerned with elucidating the role of apelin-13 analogues in cardiac arrhythmias. This work constitutes a conceptual continuation of previous reports from the research group. However, the methodological description is not clear, and it is feared that in this form it may prevent other researchers from replicating the experiment. The paper has the merits of innovation, but it is submitted that it needs a thorough rewrite.
Line 7 - why use the number 1 in the affiliation when all authors are assigned to this unit?
Line 44 - the authors did not provide in the introduction the very important information that apelin-13 has a modulatory role in endocrine interactions dictating metabolic homeostasis (10.1016/j.imbio.2021.152135, 10.2478/jvetres-2024-0042, 10.1016/j.jchemneu.2022.102171). I am, of course, aware that the present work deals with cardiac arrhythmias (as in Brugada syndrome), but for a complete view of the role of apelin-13 I find it necessary to add a paragraph and include these publications, in such a way as to briefly address these properties.
Line 75 - please avoid presenting research results already obtained. Instead, the purpose of the research should be presented.
Line 76 - In addition to the purpose of the research, please also present what the research hypothesis is and how this hypothesis was tested.
Line 83 - The methodological description is very chaotic and does not facilitate the understanding of the experiment. Please divide it into more clear subsections.
Line 85 - please state the number N of animals
Line 89 - what do the authors mean by “tissues”? Do they mean organs?
Line 98 - please provide the ethics committee approval number for this experiment
Line 166 - the description of the data analysis should not include the tool but the method.
Line 206 - the description of the statistical analysis is insufficient. Was a normal distribution tested and how? Was the ANOVA one factor or two factor? If two what factors were taken into account. Tukey test is a post hoc test.
Author Response
Line 7 - why use the number 1 in the affiliation when all authors are assigned to this unit?
The authors belong to different units within the faculty. We corrected as requested since they all also belong to the same head unit. Institute of Pharmacology of Sherbrooke Université de Sherbrooke, Sherbrooke QC. Canada.
Line 44 - the authors did not provide in the introduction the very important information that apelin-13 has a modulatory role in endocrine interactions dictating metabolic homeostasis (10.1016/j.imbio.2021.152135, 10.2478/jvetres-2024-0042, 10.1016/j.jchemneu.2022.102171). I am, of course, aware that the present work deals with cardiac arrhythmias (as in Brugada syndrome), but for a complete view of the role of apelin-13 I find it necessary to add a paragraph and include these publications, in such a way as to briefly address these properties.
A line referencing the endocrine findings was added at the beginning of the introduction as requested
Line 75 - please avoid presenting research results already obtained. Instead, the purpose of the research should be presented.
Line 75 was deleted. However, we feel that it is important to justify why we created the macrocyclic compounds. This was asked in a previous submission.
Line 76 - In addition to the purpose of the research, please also present what the research hypothesis is and how this hypothesis was tested.
Although this comment was already addressed in line 55 to 65, we nonetheless added a paragraph in line 76 to more clearly establish the goal of the study. ‘’ Our goal in this study was to determine if this new compound could alter cardiac electrophysiology. Given the known inotropic effects of Apelin we tested the effect of 2Nal on the sodium and potassium current ITo in rat cardiomyocytes.’’
Line 83 - The methodological description is very chaotic and does not facilitate the understanding of the experiment. Please divide it into more clear subsections.
Methodological section was divided as requested. Although we gave reference to a more detailed description of the methods, we nonetheless detailed the methods more extensively.
Line 85 - please state the number N of animals
N was added where needed (Fig legends)
Line 89 - what do the authors mean by “tissues”? Do they mean organs?
The word Tissues was deleted and the sentence more clearly written. We meant the cardiac ventricular tissue.
Line 98 - please provide the ethics committee approval number for this experiment
Protocol number was already provided at line 335
Line 166 - the description of the data analysis should not include the tool but the method.
It is not clear to us what this reviewer is requesting since all the information is already in the manuscript Tools: Clampex provide a way to measure data points from the recordings. We describe from lines 178 to 216, how the data were fitted to various curves. We also mention the Statistical analysis. Each figure indicates what was measured on the recordings and where. It would be redundant to specify in the methods information that is already in the figure legends. The new organisation of the methods section should alleviate this concern.
Line 206 - the description of the statistical analysis is insufficient. Was a normal distribution tested and how? Was the ANOVA one factor or two factor? If two what factors were considered. Tukey test is a post hoc test.
Normal distribution tests are used in large sample. This is not the case for our data. Moreover, all the events we measured are independent of each other. The occurrence of one event does not affect the probability another event will occur. The average rate (events per time) is constant. Two events cannot occur at the same time. This is typical of a Poisson distribution. Therefore, the statistical analysis we used based on a t-distribution with unknown population standard deviation and small sample size is the right one to use. It is common practice to compare this type of data using Student’s T or Tukey test. In our conditions, a single sided ANOVA mathematically resumes to a Student’s T test.
Tukey test was chosen because post-hoc analysis helps to draw stronger conclusions and reduce the risk of possible errors. That’s the main strength of a post-hoc test.
We added a line (216) to state that our data follow a Poisson distribution table.
Reviewer 3 Report
Comments and Suggestions for Authors
In this manuscript by Contreras-Vite et al, the authors investigate the effects of apelin derivative (2Nal) on cardiomyocyte action potentials, and rapid Na+ (INa) and transient K+ currents (Ito). The presented results demonstrate that 2Nal partially inhibits Ito current thus increasing ventricular action potential duration. It is suggested that selective inhibition of transient K+ current in cardiomyocytes by 2Nal can be used for treating cardiac arrhythmias in conditions like Brugada syndrome. These results are potentially significant; however, the manuscript does not address several important questions: What is the mechanism of 2Nal action on Ito? Is the block due to 2Nal direct binding to the channel or it works through APJ receptor, as apelin? Is 2Nal an APJ or any other GPCR agonist? If it is, how the effects of signaling downstream of APJ (or other receptors) can be separated from the direct effects of 2Nal on cardiac action potentials?
Specific comments:
1. lines 141-145. The composition of the intracellular and extracellular solutions suggest that the reversal potential for Na+ current should be negative (intracellular Na+ concentration is higher), but on Fig 5B, it is zero. Please check the solutions.
2. lines 168-169. “Cell membrane resting potential was kept at -80mV in current-clamp mode (I=0)”. It can’t be both “kept at -80 mV” and “I=0”. It is either “I” was set to zero and membrane potential was at its natural value, or membrane potential was kept at -80 mV by current injection.
3. What is the effect of 2Nal on hERG channels? Stronger block of the sustained K+ current by 2Nal may be due to hERG inhibition.
Author Response
- lines 141-145. The composition of the intracellular and extracellular solutions suggest that the reversal potential for Na+ current should be negative (intracellular Na+ concentration is higher), but on Fig 5B, it is zero. Please check the solutions.
True, one thing to consider is that upon pH adjustment the final Na+ concentration in solutions is such that the sodium gradient yielded a reversal potential of 0mV for INa. We added a sentence on line 157 to avoid that confusion.
- lines 168-169. “Cell membrane resting potential was kept at -80mV in current-clamp mode (I=0)”. It can’t be both “kept at -80 mV” and “I=0”. It is either “I” was set to zero and membrane potential was at its natural value, or membrane potential was kept at -80 mV by current injection.
We are grateful for this reviewer to notice that. Indeed, (I=0) was a typo. Voltage membrane was kept à -80 mV by current injection. We corrected the mistake.
- What is the effect of 2Nal on hERG channels? Stronger block of the sustained K+ current by 2Nal may be due to hERG inhibition.
We tested that hypothesis before doing the experiments on ITo. In our control conditions rat ERG current was roughly 5 orders of magnitude smaller than ITo and quickly ran down to negligible amplitude after 5 min under whole cell. Addition of E4031 a known blocker of HERG did not have any significant effect on our recordings with 2Nal (Not shown). But more importantly, IKr current displays a characteristic dome shaped current-voltage relationship with a peak around 0 mV and a strong rectification for more positive test potentials, leading to almost zero current at 40mV. If one looks carefully at figures 2 and 4, none of our current traces or I/V and inactivation curves show the sign of rectification expected from of a ERG contribution. We are confident that our recordings of ITo reflect specific blockade of this current by 2Nal.
Round 2
Reviewer 1 Report
Comments and Suggestions for Authors
The authors have addressed all my points!
Reviewer 2 Report
Comments and Suggestions for Authors
the authors have provided constructive responses.
Reviewer 3 Report
Comments and Suggestions for Authors
The authors have adequately addressed the reviewer’s comments.